# A Human and Rhesus Macaque Interferon-Stimulated Gene Screen Shows That Over-Expression of *ARHGEF3/XPLN* Inhibits Replication of Hepatitis C Virus and Other Flavivirids

**DOI:** 10.3390/v14081655

**Published:** 2022-07-28

**Authors:** Connor G. G. Bamford, Elihu Aranday-Cortes, Ricardo Sanchez-Velazquez, Catrina Mullan, Alain Kohl, Arvind H. Patel, Sam J. Wilson, John McLauchlan

**Affiliations:** 1MRC-University of Glasgow Centre for Virus Research, 464 Bearsden Rd, Bearsden, Glasgow G61 1QH, UK; c.bamford@qub.ac.uk (C.G.G.B.); elihu.aranday-cortes@glasgow.ac.uk (E.A.-C.); ricardo.sanchez@biontech.de (R.S.-V.); c.mullan.1@research.gla.ac.uk (C.M.); alain.kohl@glasgow.ac.uk (A.K.); arvind.patel@glasgow.ac.uk (A.H.P.); sam.wilson@glasgow.ac.uk (S.J.W.); 2Wellcome-Wolfson Institute for Experimental Medicine, Queen’s University Belfast, Belfast BT7 1NN, UK; 3BioNTech SE, 55131 Mainz, Germany

**Keywords:** Hepatitis C Virus, HCV, interferon, interferon stimulated gene, ISG, ARHGEF3, XPLN, *Flaviviridae*

## Abstract

Natural hepatitis C virus (HCV) infection is restricted to humans, whereas other primates such as rhesus macaques are non-permissive for infection. To identify human and rhesus macaque genes that differ or share the ability to inhibit HCV replication, we conducted a medium-throughput screen of lentivirus-expressed host genes that disrupt replication of HCV subgenomic replicon RNA expressing secreted *Gaussia* luciferase. A combined total of >800 interferon-stimulated genes (ISGs) were screened. Our findings confirmed established anti-HCV ISGs, such as *IRF1*, *PKR* and *DDX60*. Novel species–specific inhibitors were also identified and independently validated. Using a cell-based system that recapitulates productive HCV infection, we identified that over-expression of the ‘Rho Guanine Nucleotide Exchange Factor 3’ gene (*ARHGEF3*) from both species inhibits full-length virus replication. Additionally, replication of two mosquito-borne flaviviruses, yellow fever virus (YFV) and Zika virus (ZIKV), were also reduced in cell lines over-expressing *ARHGEF3* compared to controls. In conclusion, we ascribe novel antiviral activity to the cellular gene *ARHGEF3* that inhibits replication of HCV and other important human viral pathogens belonging to the *Flaviviridae*, and which is conserved between humans and rhesus macaques.

## 1. Introduction

The Hepatitis C virus (HCV) is the prototypic member of the genus *Hepacivirus*—in the family *Flaviviridae*, alongside yellow fever virus (YFV) and Zika virus (ZIKV), which are in the genus *Flavivirus*—that now includes viruses identified in diverse mammals, such as rodents, bats and horses [1]. HCV and other hepaciviruses can efficiently establish long-term chronic infection in the liver [2]. In the case of HCV infection in humans, chronic inflammation, cirrhosis and liver failure are typical sequelae that arise over a period of decades [3]. However, natural HCV infection is restricted to humans, although chimpanzees can recapitulate aspects of HCV pathogenesis and immunity following experimental infection [4]. Other non-human primates such as rhesus macaques are not permissive for HCV infection [5,6]. Understanding the factors underlying the species tropism of HCV is of major importance in order to aid development of better animal models for infection, which could be used to understand pathogenesis or determine vaccine efficacy [7].

Host cells have evolved numerous defense mechanisms to block viral infection and limit harmful consequences, while differences in the innate immune systems of species can underlie differential species tropism of viruses [8]. In vertebrates, a multitude of antiviral genes are induced through activation of pathogen recognition receptors (PRRs) and synthesis of three families (types I, II and III) of cytokines called interferons (IFNs) [9]. Autocrine and paracrine signaling via IFN activation of cell surface receptor complexes induces the expression of hundreds of genes, so-called ‘IFN-stimulated genes’ (ISGs), which may have direct or indirect antiviral activity, regulate IFN signaling or control inflammation [10]. Given the impact that IFNs and ISGs have on virus biology, viruses encode numerous mechanisms to evade or inhibit this response either at the point of initial detection of infection, blocking of signaling or evasion of specific antiviral ISGs [9].

HCV infection induces IFN production and signaling in vivo during acute and chronic phases of infection, which can be examined in in vitro model cell culture systems [11,12,13,14]. Furthermore, until recently, recombinant type I IFN—which demonstrates antiviral activity in vitro [15]—was used to treat chronic HCV infection [16]. Given the obvious biomedical interest in understanding how IFN controls HCV infection, numerous reports have identified a plethora of anti-HCV ISGs over the last two decades (reviewed in [17]), such as *PKR* [18,19], *MxA* [19], *IRF9* [19], *IFI6/IFI6-16* [20,21,22], *IFI27* [19], *2′5-OAS*, *IRF1* [23], *IFITM1* [24,25] and *IFITM3* [26], *MDA5* [23], *RIGI* [23], *OASL* [23], *MAP3K14* [23], *IFI44L* [23], *NT5C3* [23], *DDIT4* [23], *SSBP3* [23], *ISG56/IFIT1* [24] and *IFIT3* [27], *PLSCR1* [27], *TRIM14* [27], *NOS2* [27], *RSAD2/VIPERIN* [18,28], *ISG20* [18], *GBP1* [29], *CH25H* [30], *RTP4* [31], *ISG15* [32], *C19ORF66* [33] and *DDX60L* [34]. These studies have utilized various approaches, such as HCV pseudoparticle [25], subgenomic replicon [34] and infectious virus [23] methods to recapitulate the various stages of the virus life cycle. Moreover, there has been deployment of an array of systems for identifying ISGs with anti-viral activity, including loss- or gain-of-function approaches [20,23]. To date, one of the most comprehensive and fruitful strategies employed used an arrayed lentiviral system that expressed ~400 individual human ISGs against infectious HCV, which uncovered a number of effectors with broad anti-viral activity as well as specific anti-HCV specificity [23]. Whether other anti-HCV ISGs exist in humans or non-permissive primate species like rhesus macaques against infectious or subgenomic replicon HCV systems is unknown.

In this study, we expanded the previous screens of human ISGs with potential anti-HCV activity to include a large set of rhesus macaque ISGs [35], to determine whether additional anti-HCV IFN-mediated effectors could be found with species specificity. From the potential ISGs that inhibited HCV replication when over-expressed, we also examined their activity against flaviviruses YFV and ZIKV, identifying *ARHGEF3/XPLN* as an anti-flaviviral factor conserved between humans and rhesus macaques. Further mechanistic work would ascertain how *ARHGEF3/XPLN* over-expression inhibits HCV replication, and whether this extends to endogenous *ARHGEF3*.

## 2. Materials and Methods

### 2.1. Cells

The following three continuous human cell lines were used throughout this paper: wild-type (WT) human hepatoma Huh7 cells, Huh7 cells harboring a stable HCV SGR based on the HCV gt2a JFH-1 strain expressing the neomycin resistance gene [36], and HEK-293T cells. Huh7 cells were used for determination of antiviral activity of ISGs, while the HEK-293T cells were used to produce lentiviruses expressing ISGs. All cell lines were grown in standard high-glucose DMEM 10% FCS (*v*/*v*) without antibiotics and passaged routinely in cell culture flasks, incubated with 5% CO_2_ at 37 °C. Stable SGR cells were maintained with neomycin/G418 (ThermoFisher Scientific, Waltham, MA, USA) in every second passage (1 mg/mL). Following transduction with lentiviruses, transduced cells were selected with, and maintained in, puromycin-containing media (ThermoFisher Scientific) (between 1–3 μg/mL). Cell lines were routinely assessed for mycoplasma contamination by ‘MycoAlert’ luciferase assay (Lonza, Basel, Switzerland), but no evidence of infection was identified.

### 2.2. Molecular Biology

All plasmid stocks were produced following DNA purification by standard molecular biology procedures. ISG-expressing lentivirus transfer vectors were grown for 24 h at 30 °C before purification in commercial competent bacteria (NEB10beta, New England Biolabs, Ipswich, MA, USA). Human *ARHGEF3/XPLN* cDNA was amplified from our library vectors and modified by traditional molecular biology approaches. An amino-terminal Myc epitope tag (N-EQKLISEEDL-C) was constructed by standard and overlap high-fidelity PCR (Phusion polymerase, New England Biolabs) as were various truncations and a mutation using primers: MYC(+) 5′ CTCTCTGGCCGAGAGGGCCATGGAGCAGAAACTCATCTCAGAAGAGGATCTGGTGGCGAAGGATTACCCCTTC-3′; Full-length(+) CTCTCTGGCCGAGAGGGCCATGGTGGCGAAGGATTACCC; ARHGEF3(−) 5′-TCTCTCGGCCAGAGAGGCCTCAGACGTTACTTTCACCATG-3′; delN(+) CTCTCTGGCCGAGAGGGCCATGGAGCAGAAACTCATCTCAGAAGAGGATCTGGCGATCTTTGAGCTTTCCC-3′; N-only(−) 5′-TCTCTCGGCCAGAGAGGCCTCACTCCTGACGTTTGATTTCC-3′; W440L(+) 5′-CAAACAGCAGTTGCTTAACTGTATTCGTCAAGCCAAAGAAAC-3′; W440L(−) 5′-ACAGTTAAGCAACTGCTGTTTGTTGAAAGTGTCATTGGCTTG-3′. Amplicons were cloned into a modified ‘SCRPSY’ lentivirus vector containing only two *SfiI* sites flanking the insert. *ARHGEF3/XPLN* variants were amplified with primers flanked by *SfiI* sites and cloned directly into the SCRPSY lentivirus vector linearized by *SfiI* digestion. Resulting positive clones were verified by nucleotide sequence analysis.

### 2.3. Generation of ISG Lentivirus Library

An arrayed lentivirus library containing > 800 ORFs was produced using standard protocols. Briefly, plasmids (ISG-expressing transfer, heterologous glycoprotein VSV-G and GAG/POL packaging) were co-transfected into HEK-293T cells via lipid-mediated transfection using polyethylenimine (PEI). Lentivirus-containing supernatant was harvested 2–3 days later, clarified by low-speed centrifugation, filtered using a 0.45 μm filter and aliquoted into 96 well plates before freezing at −70 °C. For validation experiments, lentivirus ISG plasmids were re-generated from banked bacterial stocks, verified by nucleotide sequence analysis, and new lentivirus preparations were produced.

### 2.4. HCV Reagents

Two HCV systems were used for the generation of replication-competent HCV; HCV transient SGR containing the GLuc reporter based on the gt2a JFH1 strain [32] and the HCVcc gt2a Jc-1 system [37]. Infectious HCV SGR and HCVcc RNAs were generated using standard protocols [14]. Briefly, SGR or HCVcc plasmid was linearized with *Xba I* (New England Biolabs), blunt-ended using Mung bean nuclease (New England Biolabs), column purified and full-length SGR or HCVcc RNA was transcribed in vitro (T7 RiboMAX Express LargeScale RNA Production, Promega, Madison, WI, USA) followed by purification on columns and determination of quality and concentration by Nanodrop or agarose gel electrophoresis. HCV SGR RNA was transfected via lipid-mediated means or via electroporation. Lipid-mediated transfection was carried out using either PEI or lipofectamine 2000 at a ratio of 1:1 (1 μg RNA: 1 μg PEI or 1ul lipofectamine 2000) combined in Optimem 50 μL (20 min PEI, 5 min lipofectamine 2000). Huh7 cells in 96 well plates (1 × 10^4^) were transfected with RNA (200 ng/well) when subconfluent in Optimem (Thermo Fisher Scientific). Optimem/transfection mix was removed at 4 h for screening. Electroporation was carried out using standard procedures. For generation of infectious HCVcc Jc-1, protocols were carried out in a BSL3 facility and infectious virus was harvested at 3–4 days post electroporation and titrated by TCID_50_ with detection of infectious foci using an anti-NS5A antibody [38]. For infection, HCVcc-containing cultures were incubated with Huh7 cells in 12 well plates at pre-defined moi.

### 2.5. YFV and ZIKV Reporter Assays

We used two previously described recombinant reporter-expressing flaviviruses systems for a contemporary Asian strain (BeH819015 with UTR’s from PE243) ZIKV [39] and YFV [40] vaccine strain 17D. Briefly, YFV reporter virus was produced by circular polymerase extension reaction ‘CPER’ in BHK-21 cells, titrated in Huh7 cells by TCID_50_ and used for infections. Luciferase signal was measured using Nano-Glo HiBiT Lytic Detection System (Promega) in a GloMax microplate reader. ZIKV reporter virus was produced in A549 cells and titrated by TCID_50_ in Vero cells. The luciferase signal was read using Nano-Glo Luciferase Assay System (Promega) in a GloMax microplate reader. Huh7 cells expressing ISGs were challenged with the reporter ZIKV for 72 h and reporter YFV for 48 h at 37 °C, 5% CO_2_ in humidified conditions.

### 2.6. ISG Screening Protocol

Huh7 cells were seeded in 96 well plates (1 × 10^4^ cells per well) and transduced using lentivirus (75–90 μL). Two days post transduction, ISG-expressing cell lines were transfected with HCV SGR RNA for 4 h using the protocol above, followed by media changed and/puromycin (3 μg/mL) added and incubated for 3 days without changing the media. At 3 days post transfection, media was removed and aliquoted into parallel plate before being read for GLuc activity using kit (Pierce Gaussia Luciferase Glow Assay kit, Thermo Fisher Scientific) using the manufacturer’s instructions and read on a GloMax microplate reader. Relative RLU was calculated based on negative controls and plate average. Hits were defined by at least a 90% decrease in RLU in the screen.

### 2.7. Determination of Transduction Efficiency

Flow cytometry was used to determine the transduction efficiency and therefore toxicity of ISGs using the SCRPSY lentivirus vector that express the RFP tag alongside ISGs when transduced/expressed in Huh7 cell cultures via quantification of RFP-expressing cells. At 2 days after transduction, cells were trypsinised, fixed in 4% PFA and used for flow cytometry using a Guava EasyCyte flow cytometer (Millipore, Burlington, MA, USA).

### 2.8. RT-qPCR

RT-qPCR was used to determine the level of HCV RNA and for host gene expression of ISG levels using previously described protocols with the 2^−ΔΔCt^ method [14]. Total RNA was isolated from cell cultures and extracted using an RNeasy kit (Qiagen, Hilden, Germany), and cDNA was synthesized using the Applied Biosystems high-capacity cDNA kit. RT-qPCR was carried out using Taqman protocols for primers and probes against HCV (in-house [14]) as well as human *Mx1* (Hs00895608), *ARHGEF3* (Hs04241552), and the control *GAPDH* (Hs402869); assays were performed with TaqMan Fast Universal PCR Master Mix (Applied Biosystems, Waltham, MA, USA).

### 2.9. Immunoblotting

Immunoblotting was carried out on lysates from transduced cells, typically from a confluent 6 or 12-well plate (0.5–1.0 × 10^6^ cells). Lysates from cell cultures were generated using RIPA buffer extraction using manufacturer’s instructions following a PBS wash. Lysates were clarified by low-speed centrifugation and frozen at −20 °C. For running by polyacrylamide gel electrophoresis under reducing conditions (SDS-PAGE), lysates (~10 μL) were boiled along with running buffer containing beta-mercaptoethanol (Pierce Lane Marker Reducing Sample Buffer, Life Technologies, Carlsbad, CA, USA). Boiled lysates were then transferred to a hand-cast 10–12% polyacrylamide gel and ran for 2–3 h at room temperature at 100 volts. Separated polypeptides were transferred to a nitrocellulose membrane using ‘wet transfer’ conditions for 1.5 h. Membranes were blocked in 50% FCS/blocking solution and probed using primary antibodies (1:1000 dilution of beta tubulin and MYC tag antibodies (Abcam, Cambridge, UK), containing 50% FCS and PBS-T) overnight at 4 °C. Membranes were washed in PBS-T. Primary antibodies binding was visualized using corresponding secondary antibodies (Licor, Lincoln, NE, USA) and detected using the Licor imaging technology.

### 2.10. Bioinformatic Analysis of ISG Expression in HCV-Infected Human Livers

The level of RNA (fragments per kilobase of transcript per million mapped reads; FPKM) encoding putative anti-HCV ISGs was quantified from published RNAseq data [38] produced from liver biopsies of infected (HCV gt1 and gt3 (each *n* = 5)) and uninfected human livers (*n* = 4).

### 2.11. Statistical Analyses and Reproducibility

We employed extensive reproduction and validation experiments to identify ARHGEF3 as an inhibitor of HCV replication when over-expressed, including exploiting distinct but complementary model systems and approaches. Additionally, for the last figure of Section 3.4, statistical analyses (unpaired Student’s *t* test) are used, comparing data from EMPTY and ARHGEF3 WT conditions and the *p* values indicated.

## 3. Results

### 3.1. Screening of Human and Rhesus Macaque ISGs against HCV Sub-Genomic Replicon Replication

To determine whether ISGs from rhesus macaques could block HCV replication, and in an attempt to identify additional anti-HCV human ISGs, we established an ISG screening assay to systematically probe the anti-HCV activity of ISGs in the context of transient HCV RNA replication by sub-genomic replicons (SGR). The sensitivity of the assay was enhanced by deploying a system that used transiently-transfected HCV genotype (gt)2a JFH1-based replicon RNA expressing the secreted reporter protein, *Gaussia* luciferase (GLuc), in “wild-type” Huh7 cells [32]. We used previously-described libraries of human and rhesus macaque ISGs encoded by lentiviral vectors, which co-express the ISG alongside the puromycin resistance gene and red fluorescent protein (RFP) [23,35]. Huh7 cells in 96 well plates were transduced with the lentivirus libraries and subsequently transfected with HCV SGR RNA using liposome-mediated methods. Cells were selected for puromycin resistance and extracellular RLU was measured at 72 h post (hp) transfection (Figure 1A). The *IRF1* ISG positive control demonstrated robust inhibition of reporter luciferase activity comparable to values obtained from mock transduced cells (Appendix A).

Applying the above approach, a medium-throughput screen was performed with the human and rhesus macaque ISG libraries containing ~870 ISG ORFs against the luciferase-expressing transient HCV SGR. Specifically, this combined library contains 752 independent ISGs, including 493 unique ISGs (275 shared, 252 human-specific and 69 macaque-specific). Initial analysis using an arbitrary cut-off for hits as defined by >10-fold reduction in reporter activity as compared to controls revealed 54 inhibitory human ORFs (Figure 1B). This group included known anti-HCV ISGs such as *IRF1* (as described above), *RIG-I*, *RSAD2/VIPERIN* and *OASL*. Using the same criteria, the screen identified 46 inhibitory rhesus macaque ORFs (Figure 1B), including known anti-HCV ISGs (and their rhesus macaque orthologues where applicable) such as *IRF7*, *PKR* and *DDX60L*. Together, our human and rhesus macaque screens therefore identified 100 candidate antiviral effectors, the majority of which were potential novel ISG inhibitors of HCV RNA replication, such as human *ARHGEF3* and *TRIM34*, or rhesus macaque *PHF11* and *BTN3A1* (Figure 1B). There were also ISG ORFs that enhanced HCV reporter expression (Figure 1B), including both human and rhesus macaque orthologues for two genes, *CYTH1* and *HPSE*.

Before further validation of these hits, we refined the ISGs revealed by the screen. Firstly, *APOBEC* genes (e.g., *APOBEC3G*) were omitted for further study due to their known ability to restrict lentiviral transduction and likelihood thus to identify false positives [35]. Additionally, the putative rhesus macaque ISG, ‘*HLA-X*’, was excluded as it was not possible to verify that this allele was authentic due to possible generation of chimeric amplicons via PCR, which would make downstream validation challenging [41]. Reductions in HCV reporter activity could be caused by off-target effects such as activation of a global antiviral state via the induction of the IFN pathway. Therefore, to identify putative HCV restriction factors, we filtered potential hits by removing those that activate the IFN-beta promoter and interferon stimulated response element (ISRE) (>2-fold) using findings from a previously derived dataset [35]. Together, this additional filtering resulted in a final selection of 62 ISGs (32 human and 30 macaque), which we proceeded to validate.

### 3.2. Validation of ISG Inhibitors of HCV SGR Replication

All potential inhibitory hits (alongside *CYTH1* and *HPSE* that enhanced replication) from the primary screen were tested in an independent secondary validation screen since it is possible that the initial screen could identify false positives other than those that we had excluded. Importantly, this secondary screen was conducted using independently generated lentivirus stocks following plasmid confirmation of gene identity. The same experimental procedure as described above was employed except data were obtained in triplicate. In these experiments, we took a ~2-fold inhibition of the mean values in RLU compared to the ‘no ISG control’ of empty/EGFP-expressing lentivirus transductions to select positive hits (Figure 2A,B). Of the ISGs assessed, about 50% were validated as inhibitory in the secondary screen. In parallel, the transduction efficiency of the lentivirus stocks was also determined by measurement of the number RFP-positive cells using non-puromycin-selected samples by flow cytometry. While 8 ISGs met the inhibition criteria, they exhibited poor transduction efficiency and were excluded as likely false positives (e.g., human *SAMD9L*, *NOS2A & PARP10*, and rhesus macaque *MLKL*, *C5ORF39* and *BST2/TETHERIN* (Appendix A). These genes were presumed to be either cytotoxic or possess anti-lentiviral vector activity.

Excluding known inhibitory human ISGs (*RNASEL*, *DDX58/RIGI*, *IRF1*, *DDX60*, *OASL* and *ADAR* as well as rhesus macaque *RNASEL*, *IFNB1*, *IRF1* and *DDX60L*), we identified the following ISGs as novel inhibitors of HCV SGR replication: human *ARHGAP17*, *ARHGEF3*, *MICB*, *CDKN1A*, *SRBII*, *FAM46C*, *TRIM34* and *SLC15A3*, and rhesus macaque *NCOA7*, *COMMD3*, *SRBII*, *CASP1*, *PHF11*, *PTAR1* and *BTN3A1*. Human *SLC15A3* was taken forward as, although it did not reach 2-fold inhibition, it had a strong effect in our experiments compared to the well-established restriction factor *RSAD2/VIPERIN*. Furthermore, we detected inhibition by two isoforms of the rhesus macaque orthologue to human *NCOA7*, *PTAR1* and *BTN3A1*; only the longest isoform of each was examined further. One ISG, *SRBII*, was inhibitory in both human and rhesus macaque screens.

The capacity to restrict HCV replication by ISGs from the primary and secondary screens was further evaluated by monitoring HCV RNA replication following electroporation of SGR RNA into Huh7 cell lines that had constitutive expression of ISGs (Figure 2C–F). In these experiments, we used an arbitrary cut-off of 2-fold reduction in viral RNA to be considered inhibitory. RLU reporter measurements were conducted on the WT SGR (Figure 2C) and the replication-defective ‘GND’ mutant (Figure 2D) at 72 hp electroporation to ascertain any role of ISGs in disrupting primary translation of SGR RNA. We were able to establish stable Huh7 cell lines expressing ISGs for all genes except human *MICB*, which exhibited toxicity during long-term expression in cell culture and so was not explored further. Stable expression of most ISGs (*n* = 17) reduced expression of the GLuc reporter following electroporation compared to controls (EGFP or empty cells) except *SRBII-Mm* (Figure 2C,D). Interestingly, the ISG that previously enhanced reporter expression, *CYTH1*, did not display the same phenotype in this assay, suggesting that the gene may affect transfection of SGR RNA. From analysis of reporter expression from the GND replication-defective construct, some ISGs (*CDNK1A*, *FAM46C*, *RNASEL*, *SRBII-HS*, *PHF11*, *CASP1*) reduced translation (Figure 2D). The ratio of WT:GND reporter expression was calculated as a measure of the ability of each ISG to inhibit HCV RNA replication compared to translation of input SGR RNA (Figure 2E). By this measure, all ISGs apart from *CDKN1A* inhibited HCV RNA replication. Although we used WT:GND to exclude *CDKN1A*, we note that the biological significance of this ratio is challenging to interpret. Furthermore, the relative accumulation of HCV SGR RNA in electroporated Huh7-ISG cells was measured to identify those ISGs that could promote off-target inhibition by altering reporter expression/secretion rather than specifically interfering with HCV RNA replication. Most ISGs inhibited HCV RNA accumulation as compared to control lines that did not express any ISG, mirroring their inhibition of reporter activity except for *PTAR1*, *NCOA7* and *SRBII-Mm* (Figure 2F). Indeed, *NCOA7* and *SRBII-Mm* enhanced HCV RNA accumulation. This increase in HCV RNA accumulation by *SRBII-Mm* contrasts with the inhibitory effect of *SRBII-Hs* on viral RNA replication, which may be worthy of future study. Based on these findings, *CDKN1A*, *PTAR1*, *NCOA7* and *SRBII-Mm* were excluded from further analysis. Thus, the remaining novel ISGs that interfered with HCV RNA replication were human *ARHGAP17*, *ARHGEF3*, *FAM46C*, *SLC15A3*, *SRBII-Hs*, *TRIM34* and rhesus macaque *BTN3A1*, *CASP1*, *COMMD3* and *PHF11*.

### 3.3. Assessment of Inhibitor Activity against Replication of Infectious Virus

As replicon-based assays do not represent the complete HCV life cycle, we screened the final selected 10 Huh7-ISGs as stable cell lines using HCVcc Jc1 full-length virus and measured viral RNA accumulation at 72 h post infection (hpi) (Figure 3). For these assays, we included *IRF1* and *RSAD2/VIPERIN* as additional controls, since these ISGs inhibit RNA replication in both the HCV SGR (Figure 2) and infectious systems. In this experiment, *IRF1* and *RSAD2/VIPERIN* overexpression reduced HCV RNA accumulation by >100 and ~10-fold respectively (Figure 3). Consistent with previous experiments, in this assay, we also used 2-fold cut off as a criterion for an ISG to have inhibitory activity. Only *ARHGEF3* achieved this threshold, although other ISGs such as *FAM46C* and *TRIM34* did have an effect albeit to a less dramatic extent. Interestingly, in contrast to replicon-based assays, *ARHGAP17* now enhanced HCV RNA accumulation by ~2.5-fold. ISGs exhibiting contradictory effects in the different challenge models (SGR transient versus full-length virus) likely reflects intrinsic differences in respective model systems (e.g., gene content, delivery system, etc.).

### 3.4. ARHGEF3 Inhibits HCV Replication

As *ARHGEF3* (also known as *XPLN*) gave the most robust inhibition of HCV replication, including in the infectious virus system, and its role in HCV infection has not been previously examined, we chose to study it in greater detail. Although only human *ARHGEF3* was identified in our initial screen, the rhesus macaque ISG library also contained *ARHGEF3*, although it did not give any apparent antiviral effect in the primary screen. We then examined the capacity of *ARHGEF3* to inhibit RNA replication in both the SGR and infectious HCV systems using Huh7 cells that stably expressed the rhesus macaque orthologue. In both systems, the human and rhesus macaque orthologues inhibited RNA replication to similar levels, demonstrating conservation of anti-HCV (SGR and HCVcc) activity for *ARHGEF3* in both species (Figure 4A,B).

We next examined the role and potential mechanism of *ARHGEF3*-mediated inhibition of HCV RNA replication. Hence, three *ARHGEF3* mutants from the human orthologue were generated with an N-terminal MYC-tag to facilitate comparative analysis of their abundance. *ARHGEF3* has two major established activities, acting as a guanine nucleotide exchange factor (GEF) for various Rho proteins [42], and activation of mammalian target of rapamycin complex 2 (mTORC2) [43]; these activities are encoded by distinct regions of the protein. RhoGEF activity can be blocked by single point mutations in the GEF domain while mTORC2 inhibition is controlled by the N-terminal region (positions 1–125) of the protein [39]. To test whether either of these functions contributed to the antiviral activity of *ARHGEF3*, we made a single RhoGEF activity ablation point mutant (W440L hereafter termed ‘delGEF’), an N-terminal truncation (positions 1–125) of *ARHGEF3* (termed ‘delN’) and expressed only the N-terminal fragment (positions 1–125) of *ARHGEF3* (termed ‘N’). We generated stable Huh7 cell lines for the WT, mutated and deleted forms of *ARHGEF3*, monitored protein expression using the Myc tag (Appendix A) and quantified HCV RNA replication by measuring Gaussian luciferase activity using the SGR assay (Figure 4C). The WT version of Myc-tagged *ARHGEF3* inhibited HCV RNA replication to a similar degree as the non-tagged WT form of the gene with the HCV SGR system (Figure 4B). By contrast, antiviral activity was ablated for all three variants of *ARHGEF3* (Figure 4C). This loss in activity was not due to any major differences in protein levels between the WT and mutants (Appendix A). These data demonstrate that the anti-HCV activity of *ARHGEF3* is dependent on functions across the entire gene including both the N-terminal domain, which is responsible for mTORC2 activation, and RhoGEF activity. However, the mTORC2-sufficient N domain failed to demonstrate antiviral activity, although it gave reduced expression compared to the others (Appendix A)

Given their relatedness, we hypothesized that the antiviral activity of *ARHGEF3* may extend to flaviviruses as well as HCV. To test this hypothesis, we challenged stable cell lines expressing *ARHGEF3* with yellow fever virus (YFV; derived from vaccine 17D) and Zika virus (ZIKV), two mosquito-borne flaviviruses (Figure 5A,B). The viruses chosen for these experiments were recombinant strains that expressed the nanoluc or HiBiT reporters, similar to the luciferase-expressing HCV systems used herein [39,40]. Reporter activity following infection with both viruses was decreased in *ARHGEF3*-expressing Huh7 cells although not to the same extent as the positive control *IRF1*. YFV replication was inhibited to a greater extent (5-fold) compared to ZIKV (2-fold). These data suggest that *ARHGEF3* may exert antiviral effects not only on HCV but other members of the *Flaviviridae*.

### 3.5. ARHGEF3 Is Expressed and Modestly Up-Regulated in the Human Liver during Chronic HCV Infection

Given the observed inhibition of HCV replication by *ARHGEF3* following infection of Huh7 human hepatoma cells, we wanted to determine whether the gene was expressed in chronically HCV-infected liver in vivo where it could also contribute to controlling viral replication. By analyzing RNAseq data from HCV gt1- and gt3-infected livers [44], we compared the levels of *ARHGEF3* expression in uninfected and infected liver biopsies along with a number of other anti-HCV ISGs (Figure 6A,B) These data showed that *ARHGEF3* transcription was detected in uninfected liver, albeit at a very low level compared to other ISGs such as *IFITM3* (Figure 6A). In HCV-infected liver biopsies, *ARHGEF3* was modestly upregulated 1.5–2-fold during HCV gt1 and gt3 infection; by comparison, *IFI27* and *RSAD2/VIPERIN* were induced >10–100 fold during HCV infection (Figure 6B). In vitro ISG induction by IFN-alpha and IFN Lambda3 stimulation in Huh7 cells yielded negligible induction of *ARHGEF3* compared to *MX1* (Appendix A). Thus, *ARHGEF3* expression could be detected in the liver or liver-derived cells where it may be very modestly induced during infection or IFN stimulation compared with other anti-HCV ISG factors.

## 4. Discussion

The outcome of acute HCV infection is variable in natural infection of humans and is intimately linked with host immune responses, including the ability to control infection through the IFN system [16,45]. Similarly, in animal models, the course of infection and clinical pathology also differs from human infection, such as rhesus macaques which are non-permissive to infection in vivo [4,5,6]. One hypothesis is that species-specific differences in IFN-mediated immunity may contribute to the observed distinct outcomes. Since the antiviral potency of IFN is mediated through the induction of a large spectrum of core ISGs [10,23], there have been previous screening studies to identify specific ISGs that interrupt the HCV infectious cycle. The most comprehensive approach utilized a screening method similar to that adopted here but was not applied to the rhesus macaque ISG library [35] and did not utilize HCV SGR systems [23]. It should be noted that while rhesus macaques cannot be productively infected in vivo, hepatocytes cultured ex vivo from the species remain permissive [46], although the relative efficiency of HCV replication between human and rhesus macaque hepatocytes has not been compared. By way of such an example, we revealed in a recent report that rhesus macaque and chimpanzee *IFNL4* had greater antiviral activity than their human orthologue due in part to an amino acid variant at codon position 154 in the *IFNL4* gene, which may impact the outcome of HCV infection [14]. One consideration for our approach is that the antiviral activity of ISGs may be dependent on the host cell and thus, it is possible that expression of macaque ISGs in human cells could enhance or reduce their activity.

Our study has utilized both transient HCV SGR and HCVcc systems, uncovering *ARHGEF3/XPLN* as an antiviral factor conserved between humans and rhesus macaques. Notably, in our experiments, there are differing outcomes between transient SGR and HCVcc systems. We identified several potential antiviral ISGs (e.g., *TRIM34*, *FAM46C*) with the SGR approach, yet only one hit achieved our arbitrary cut-off for reducing viral RNA by >2-fold with infectious HCVcc (summarized in Appendix A). The SGR and HCVcc systems are based on the same gt2a viral strain (JFH1/Jc-1 [37,47]) and for certain experiments, the same method was employed to quantify RNA abundance. However, there are substantial biological and experimental differences between the approaches using transient SGR transfection and HCVcc infection. Firstly, SGR RNA is transfected or electroplated into Huh7 cells, unlike the authentic infectious virus. Delivery of viral RNA is therefore distinct, as is the potential to induce an IFN response from introduction of naked RNA to the cytoplasm, as we used Huh7 cells and not Huh7.5 cells, which are deficient in RIG-I-dependent sensing [48]. Additionally, by definition, the SGR system lacks genes (such as HCV-encoded capsid, envelope, p7 and NS2 genes) present in full-length HCVcc that have been shown to affect sensitivity to certain ISGs [21]. Future work may, however, interrogate the role of *TRIM34*, *FAM46C* and *SLC15A3* in inhibiting HCVcc despite the modest inhibition.

Given that our strategy employed an initial screen using transient replication with the HCV SGR approach followed by validation employing an infectious assay system, the potential anti-viral host factors would likely target either translation or replication of HCV RNA. Thus, we would not anticipate identifying host factors that would only interrupt either virus entry or assembly. Having identified novel ISGs compared to previous screens using the SGR approach, the only anti-viral factor that met our strict criteria for reducing replication in an infectious system was *ARHGEF3/XPLN*. Interestingly, the rhesus macaque orthologue showed similar activity although it was not identified in the initial library screen, which is likely due to high false negatives in initial assays. Nonetheless, this indicates conservation of function for both human and rhesus macaque orthologues. Moreover, *ARHGEF3/XPLN* displayed antiviral activity in assays for both ZIKV and YFV replication. Thus, the gene apparently has broader anti-viral activity against other positive-sense, single-stranded RNA viruses in the *Flaviviridae* family.

*ARHGEF3/XPLN* is a guanine nucleotide exchange factor, which specifically stimulates and interacts with two Rho GTPases, RhoA and RhoB, involved in cytoskeletal regulation [49]. In common with other GTPases, ARHGEF3 possesses both a Dbl homology (DH) domain, responsible for catalytic activity, and a pleckstrin homology (PH) domain [42,49]. ARHGEF3 also interacts with and inhibits mTORC2 at endogenous levels, which regulates cell development [43]. However, this function of ARHGEF3 does not require its GEF activity. ARHGEF3 protein is most highly expressed in the brain and skeletal muscle, and to a lesser extent, in heart and kidney, but expression could not be detected in the liver [49]. More recently, the mouse orthologue of ARHGEF3 has been implicated in regulating muscle regeneration in a process that involves autophagy [50]. Thus, ARHGEF3 influences a number of physiological processes, although detailed insight into all of its possible functions have not been explored. Additionally, the higher levels of ARHGEF3 expression in several tissues may partially explain the lack of robust infectivity in non-hepatic cell lines.

Our analysis permitted some preliminary mechanistic insight into *ARHGEF3/XPLN* inhibition of RNA replication for HCV, YFV and ZIKV in vitro. From our experiments with the different HCV systems available, *ARHGEF3* was able to effectively inhibit de novo RNA replication in cells transfected with SGR RNA. However, it could not exert a similar antiviral effect in cells that constitutively harbored the subgenomic replicon, although the positive control *IRF1* had muted activity with this approach also (Appendix A). Presumably, persistent HCV RNA replication may adapt the intracellular environment and regulatory processes such that certain ISGs are less effective. Thus, the factor may only function as an antiviral upon initial infection when RNA replication is being established. The ability of the rhesus macaque orthologue to also inhibit replication indicates that any antiviral activity is conserved in other species. This is not surprising given that human and rhesus macaque orthologues differ at only 3 amino acid positions (data not shown). Moreover, *ARHGEF3* variants that either introduced a mutation which blocked GEF activity or removed the region of the protein responsible for mTORC2 inhibition ablated antiviral activity. However, care must be taken in interpreting results from the N domain given its reduced expression at the protein level. Thus, it was not possible identify the function in *ARHGEF3* responsible for blocking viral RNA replication. Speculatively, it is possible that *ARHGEF3* directly targets early replication processes shared by members of the *Flaviviridae*, through limiting autophagy, which is needed for the formation of replication organelles of these viruses [51]. Further studies would require more detailed analysis of *ARHGEF3* to determine the precise mechanism for inhibiting viral RNA replication and the cellular processes that limit viral RNA synthesis. These next experiments would include specifically assessing the capability of endogenous ARHGEF3/XPLN to inhibit HCV replication. Additionally, cells deficient in ARHGEF3/XPLN could be generated and ARHGEF3 reintroduced. High resolution flurescence microscopy could be used to determine the localization of ARHGEF3 with regard to replication complexes. However, given ARHGEF3 antiviral activity, it may be challenging to identify cells co-expressing ARHGEF3 and HCV replication proteins at sufficiently high-enough levels to facilitate imaging.

The IFN system can pose a significant barrier to HCV infection in vitro and in vivo. One question resulting from our studies is whether this inhibition could be influenced by *ARHGEF3/XPLN* in the liver in natural infection. Our data show that it can be detected in the liver albeit at low levels, and we also show that it can be induced, although not to a substantial extent, by HCV infection in vivo and IFN in vitro in Huh7 cells. However, compared to other ISGs with potent anti-HCV activity and higher expression levels, the effect of *ARHGEF3/XPLN* in vivo within the liver is likely to be modest. Nevertheless, we have demonstrated that *ARHGEF3* also exhibits anti-viral activity against two other flaviviruses, YFV and ZIKV, which have significant extra-hepatic tropism, although YFV can infect hepatocytes in vivo. Thus, induction of *ARHGEF3* may have broader effects on other single-stranded viruses that can infect tissues where the gene is expressed to high levels. We propose that future studies of *ARHGEF3*, for example, using pre-existing animal models, may focus efforts on its properties as an anti-viral inhibitor that is stimulated by the IFN response.

## Figures and Tables

**Figure 1 viruses-14-01655-f001:**
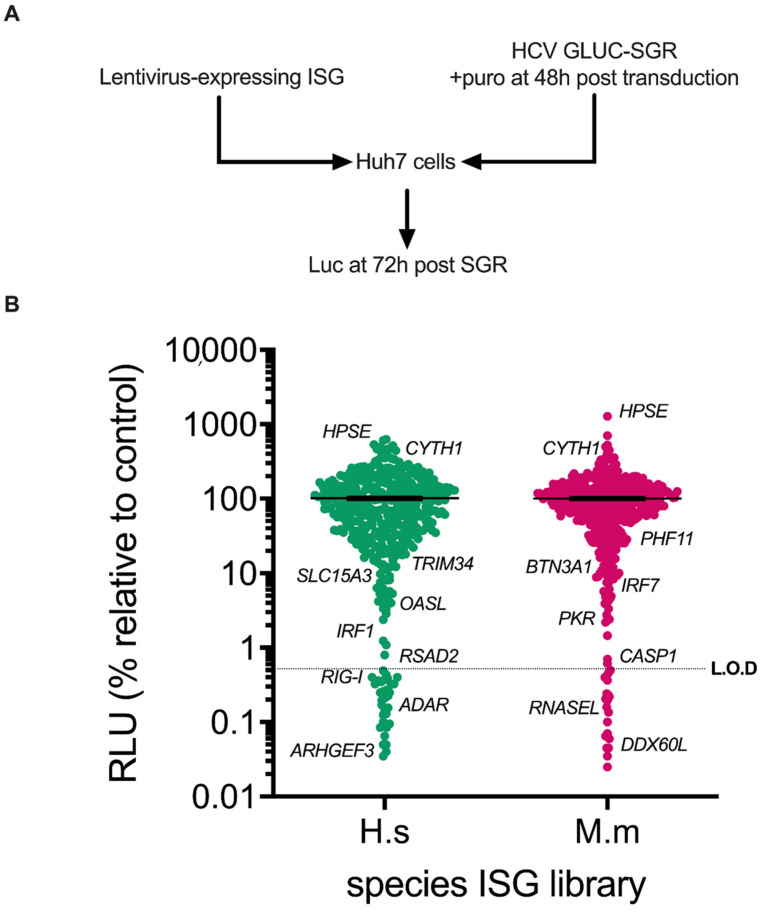
Screening for ISGs from humans and rhesus macaques that influence HCV-SGR luciferase reporter activity. (**A**) Schematic depiction of the screening protocol used in the study showing that Huh7 cells were transduced and transfected with luciferase-expressing HCV SGR after 48 h and the secreted luciferase levels were quantified at 72 h post-transfection (hpt). (**B**) Dot plot of RLU detected for ISG-expressing cells transfected with HCV-SGR RNA for human (Hs; green) and rhesus macaque (Mm; red) ISGs, relative to the control ISGs (%) from 72 hpt. The primary screen was performed as single assays. Genes of interest are marked on the figure. Limit of detection (L.O.D) highlighted with a dotted line.

**Figure 2 viruses-14-01655-f002:**
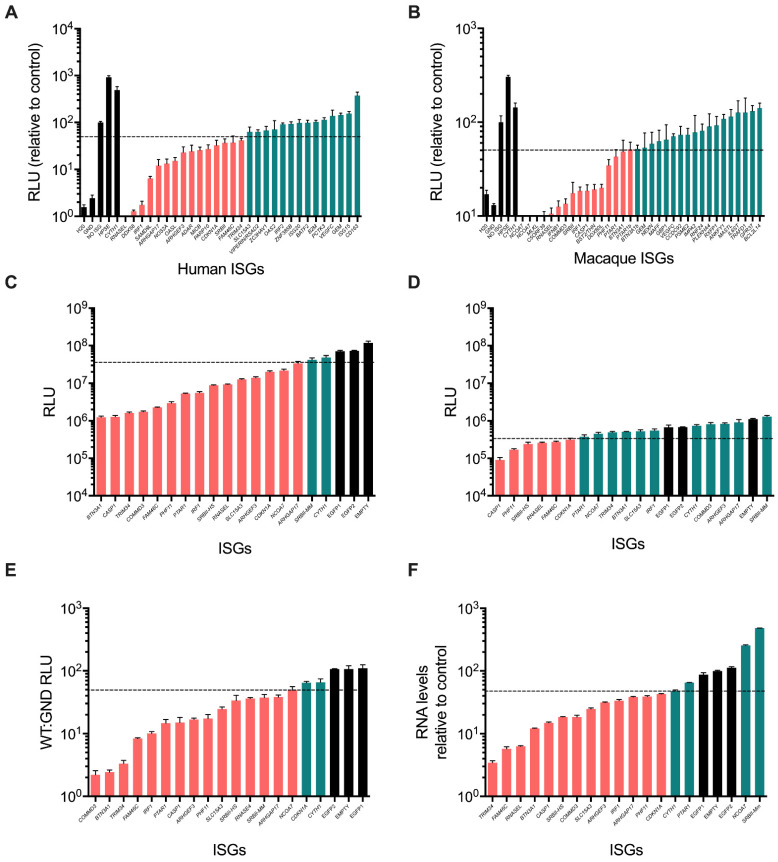
Secondary screen validation of anti-HCV-SGR ISGs from humans and rhesus macaques detected in the initial screen. (**A**,**B**) RLU detected at 72 hpt in HCV-SGR RNA (wild-type [WT]) transfected cells expressing human (**A**) or rhesus macaque (**B**) ISGs. (**C**,**D**) Validation of anti-HCV-SGR activity of human and rhesus macaque ISGs against WT (**C**), or replication-defective HCV-SGR RNA (**D**) at 72 hpt. The ratio of WT:GND RLU (**E**) and HCV RNA levels (**F**) was calculated at 72 hpt. For the data in panel F, WT HCV-SGR RNA was measured by RT-qPCR following electroporation of SGR RNA into stable puromycin-selected ISG-expressing Huh7 cell lines. Negative controls are shown (black) and ISGs that gave a >2-fold reduction in HCV RNA replication are indicated (pink). ISGs that do not achieve this threshold are shown in green. Values from triplicate wells as technical replicates were used and variation is shown as standard error of the mean.

**Figure 3 viruses-14-01655-f003:**
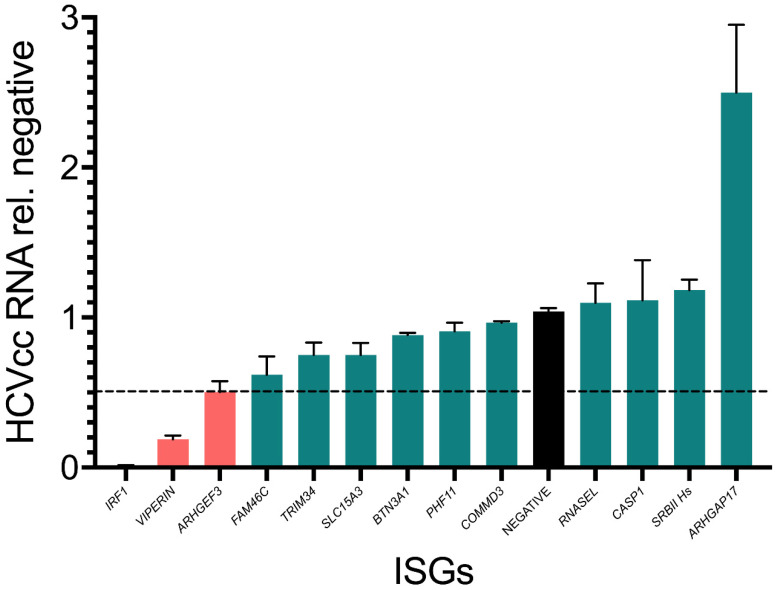
Validation of ISGs with activity against HCVcc. The effect of ISG expression on replication of full-length Jc1-HCVcc was ascertained following infection of stably transduced ISG-expressing cell lines by quantification of HCV viral RNA abundance by RT-qPCR at 72 hpi (moi = 0.1). Negative controls are shown (black) and ISGs that reached a threshold of >2-fold reduction in viral replication are shown in pink. ISGs that do not achieve this threshold are shown in green. Values from combined duplicate technical replicates from three independent experiments were used and variation is shown as standard error of the mean.

**Figure 4 viruses-14-01655-f004:**
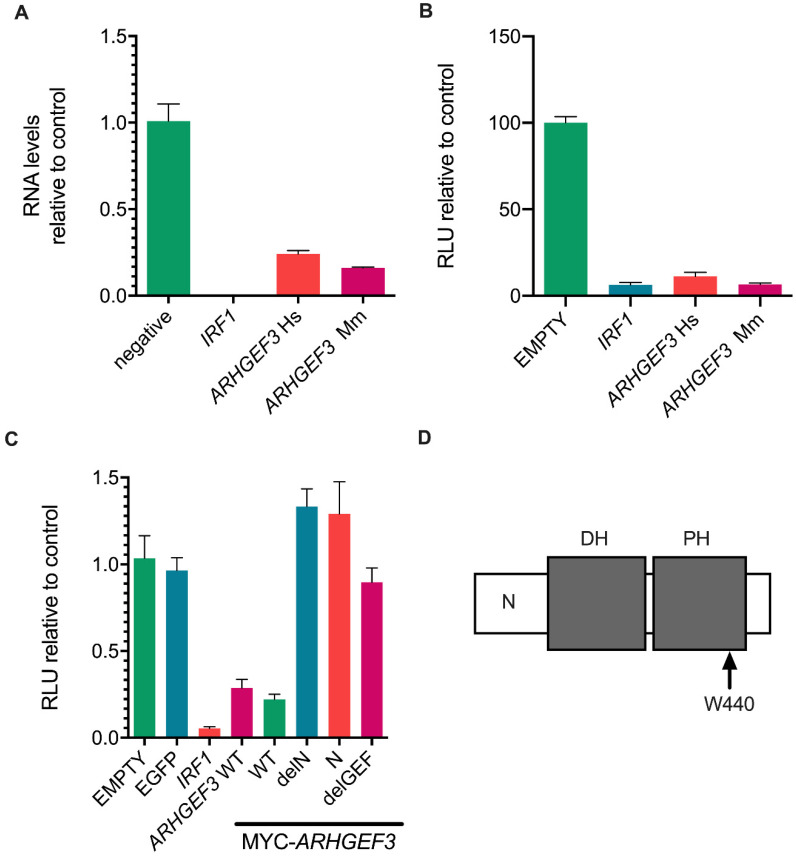
Validation of the antiviral activity of *ARHGEF3/XPLN* on HCV RNA replication. (**A**) Relative HCV RNA abundance from HCVcc infection of Huh7 cells expressing *ARHGEF3* from Human (Hs) or rhesus macaque (Mm) as compared to negative control at 72 hpi. (**B**) Relative RLU (%) following HCV SGR RNA transfection of Huh7 cells expressing *ARHGEF3* from Human (Hs) or rhesus macaque (Mm) as compared to negative control at 72 hpt. (**C**) Effect of *ARHGEF3/XPLN*-specific mutations (Myc-tagged WT, N-terminus truncation, N-terminus only and the GEF-inactivation mutant W440L) on HCV RNA replication following SGR RNA transfection. (**D**) Schematic of *ARHGEF3* showing the N (amino acids 1–125), diffuse B cell lymphoma homology (DH) and plekstrin homology (PH) domains, alongside the location of the W440 residue. RLU activity was measured and compared to controls at 72 hpt. Values from combined three technical replicates from two independent experiments were used and variation is shown as standard error of the mean.

**Figure 5 viruses-14-01655-f005:**
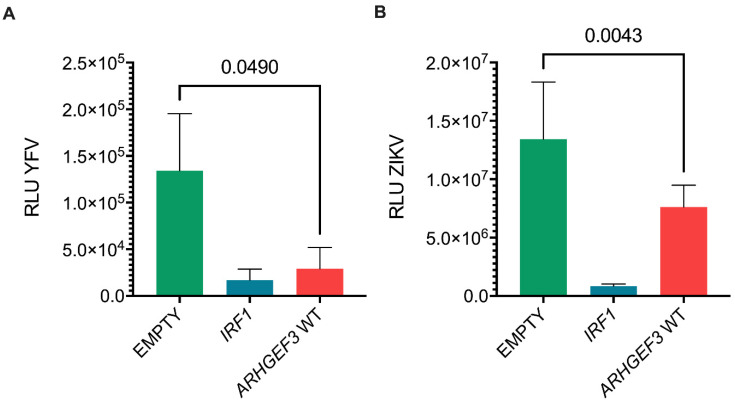
Antiviral activity of ARHGEF3 against flaviviruses. RLU activity in Huh7 cells stably expressing *ARHGEF3/XPLN* or *IRF1* following infection with recombinant luciferase-expressing YFV (**A**) and ZIKV (**B**) as measured at 48 hpi and 72 hpi, respectively. Values from 3 technical replicates for YFV and 9 technical replicates for ZIKV were used and variation is shown as standard deviation of the mean alongside *p* values for Student’s *T* test between EMPTY and *ARHGEF3* WT results.

**Figure 6 viruses-14-01655-f006:**
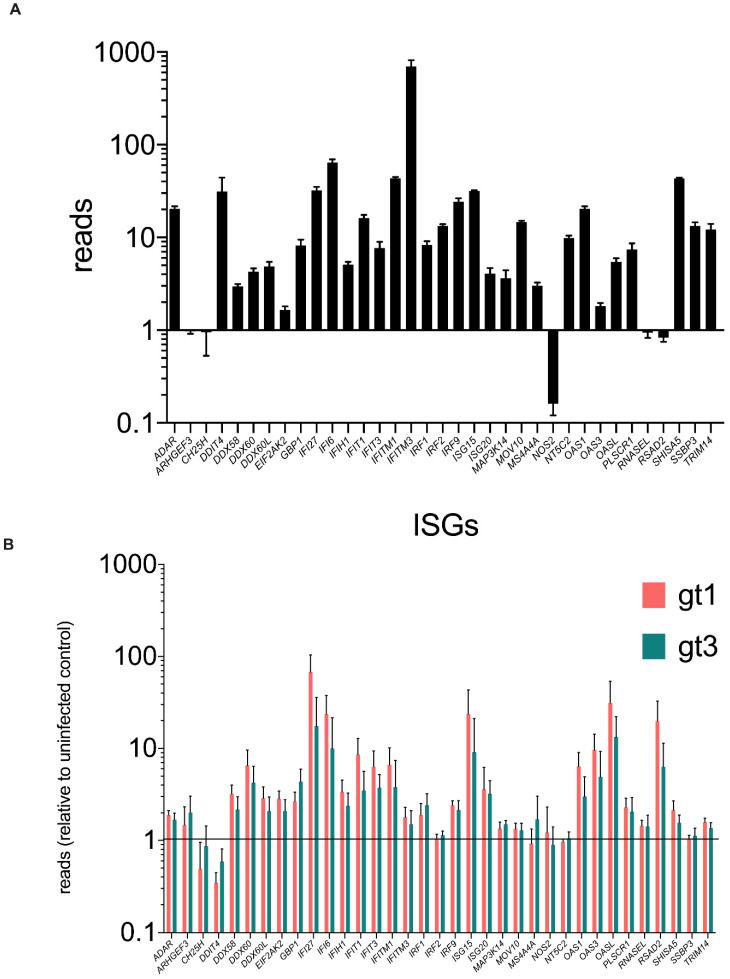
Abundance of ARHGEF3/XPLN RNA in vivo in HCV-infected human liver biopsies. RNA levels (FKPM) of known anti-HCV ISGs in liver biopsies from uninfected individuals (*n* = 4) (**A**). Fold-change in RNA (FPKM) levels of known ISGs in liver biopsies from infected individuals (HCV gt1: pink; gt3: green (each *n* = 5)) compared to uninfected controls (**B**).

## Data Availability

Not applicable.

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
