# Peer review of "A Human and Rhesus Macaque Interferon-Stimulated Gene Screen Shows That Over-Expression of ARHGEF3/XPLN Inhibits Replication of Hepatitis C Virus and Other Flavivirids"

_viruses, 2022, doi:10.3390/v14081655_

Round 1

Reviewer 1 Report

In their submitted manuscript, the authors describe their screening approach to identify novel interferon (IFN) stimulated genes (ISGs) impacting HCV replication in human and rhesus macaque. In contrast to previously published similar endeavors, their approach was primarily based on a sub-genomic reporter replicon assay instead of actual virus infection. Nonetheless, real virus infection (HCVcc) was used as a last validation step. The authors end up identifying one single new ISG that showed antiviral activity in all of their assays, which they briefly characterized further: ARHGEF3/ZPLN. They found that ZPLN from both, human and macaque, was capable to inhibit HCV replication to some degree, and this was not specific to HCV but also for related viruses from the flavivirus family (Zika and Yellow Fever virus). They could furthermore show that the antiviral activity of ZPLN required both, the GEF activity, as well as the N-terminal domain, which is more related to mTOR regulation. Lastly, they show that ZPLN is only slightly expressed in livers, but can be (very modestly) induced upon HCV infection in patient livers, most likely by IFN (although it was largely IFN-independent in cell culture).

Overall, the described work appears very solid and technically sound. The manuscript is very well written, using a clear and comprehensible language; methodology is mostly clearly described (although some aspects can be elaborated more, see below) and data is presented clearly and appealingly. Novelty and scientific impact are comparably small, but still worth reporting. 

I have very few points, that I would strongly suggest the authors to address prior publication.

1)    More technical detail required for the screening procedure: describe the origin of the ISG library– is it the famous Schoggins library (human) and the Bieniasz rhesus one? Detail more exactly in which assays electroporation of HCV SGR was used and when SGR was transfected using lipofection! Primary screen was carried out in 96-well plates, but text says electroporation was used (line 94)– how was that done?

2)    Fig 1: the authors should additionally to fig 1 provide the full data from their primary screening as an Excel table or the like (gene name and “RLU relative to control”).

3)    What’s the explanation for this huge effect size in the primary screen (cut-off set to 10-fold reduction), whereas in all downstream assays (even if otherwise technically identical, e.g. Fig 2) effects are so much smaller and the cut-off had to be reduced to 2-fold? Normally, one would rather expect the opposite (least effect size in primary)… This should be discussed explicitly. Also note, that this great effect size in primary screening makes it difficult to argue why the authors have not found Mm-ARHGEF3 in the primary screen but could “validate” it later on despite much smaller effect size overall (compare line 298f). 

4)    Fig. 2E: “replication” (RLU after 72h) comprises translation *and* replication fitness, but presumably coupled in a highly non-linear fashion. Hence, I am unsure whether the ratio WT/GND is straightforward to interpret. As its results also do not impact on the further analyses in the manuscript, I recommend removing it or moving it into the supplements.

5)    The authors should assemble a summarizing graph of all ISG candidates across all screening/validation experiments, e.g. as a heatmap showing the effect size of each ISG (coded as color) across all experiments. Like this, one might be able to identify further interesting hits (e.g. highly consistent but not overly huge effect size; contrary: large effect size in single assay, but none or even reversed effect in other assays).

6)    In general, there is very little information on the applied statistics– I recommend adding a short paragraph to materials and methods. I understand hardly any experiments have been repeated independently, but rather only technical replicates were employed? This must be stated clearly. Fig. 4 lacks a statement on replicates and statistics. Fig. 5 uses 9 and 5 replicates (technical or biological?) for the two viruses– this is suboptimal when using SEM (as n influences SEM). Fig. 5 might be better off with SD and/or showing individual measurements on top; also a t-test could be applied, especially to ARHGEF3 in ZIKV (also: why do empty controls have no error bar– please indicate variation!). 

Minor:

1)    Fig. 1: indicate that Y-axis is in “%”

2)    Line 231: “extend to other flaviviruses…” remove “other”

Author Response

Dear Editor,

Thank you for your patience with our response to reviewers given the number of points to address, and other circumstances outside of our control that prevented us from getting this to you sooner. We wish you thank you and the three kind reviewers for their helpful comments and insights into our manuscript. Firstly, we note that all three reviewers said: the work was solid and technically sound (r1), our work had interest and novelty (r2), and it was well-conducted, technically sound and convincing (r3). There were suggestions around further significant mechanistic studies and follow up experiments. However, given the time limit for resubmission and the significant amount of work suggested to be carried out, we have decided to amend our conclusions and have now altered our text, incorporating the reviewers suggestions as critical further work. We have now addressed all the comments and we hope that the manuscript is satisfactory to publish in your journal and special issue on HCV.

Reviewer 1:

In their submitted manuscript, the authors describe their screening approach to identify novel interferon (IFN) stimulated genes (ISGs) impacting HCV replication in human and rhesus macaque. In contrast to previously published similar endeavors, their approach was primarily based on a sub-genomic reporter replicon assay instead of actual virus infection. Nonetheless, real virus infection (HCVcc) was used as a last validation step. The authors end up identifying one single new ISG that showed antiviral activity in all of their assays, which they briefly characterized further: ARHGEF3/ZPLN. They found that ZPLN from both, human and macaque, was capable to inhibit HCV replication to some degree, and this was not specific to HCV but also for related viruses from the flavivirus family (Zika and Yellow Fever virus). They could furthermore show that the antiviral activity of ZPLN required both, the GEF activity, as well as the N-terminal domain, which is more related to mTOR regulation. Lastly, they show that ZPLN is only slightly expressed in livers, but can be (very modestly) induced upon HCV infection in patient livers, most likely by IFN (although it was largely IFN-independent in cell culture).

Overall, the described work appears very solid and technically sound. The manuscript is very well written, using a clear and comprehensible language; methodology is mostly clearly described (although some aspects can be elaborated more, see below) and data is presented clearly and appealingly. Novelty and scientific impact are comparably small, but still worth reporting. 

I have very few points, that I would strongly suggest the authors to address prior publication.

1)   More technical detail required for the screening procedure: describe the origin of the ISG library– is it the famous Schoggins library (human) and the Bieniasz rhesus one? Detail more exactly in which assays electroporation of HCV SGR was used and when SGR was transfected using lipofection! Primary screen was carried out in 96-well plates, but text says electroporation was used (line 94)– how was that done?

We thank the reviewer for these points regarding methodological reporting. We have now made it explicit as to the origin of both ISG libraries in the text (line 98). We can confirm that they are the Schoggins et al., and Kane et al., libraries.

The reference to electroporation in line 103 is a mistake and we apologise for any confusion; it should have read transfection. Both electroporation and liposome-mediated transfection were carried out. For most assays delivery of HCV replicon RNA was achieved using liposomes while in figures 4 C-F it is electroporation.

2)    Fig 1: the authors should additionally to fig 1 provide the full data from their primary screening as an Excel table or the like (gene name and “RLU relative to control”).

We have now included this as a supplementary file 1.

3)    What’s the explanation for this huge effect size in the primary screen (cut-off set to 10-fold reduction), whereas in all downstream assays (even if otherwise technically identical, e.g. Fig 2) effects are so much smaller and the cut-off had to be reduced to 2-fold? Normally, one would rather expect the opposite (least effect size in primary)… This should be discussed explicitly. Also note, that this great effect size in primary screening makes it difficult to argue why the authors have not found Mm-ARHGEF3 in the primary screen but could “validate” it later on despite much smaller effect size overall (compare line 298f). 

We thank the reviewer for this insightful comment. While we do not know precisely why this is the case, it is likely due to the variable nature of the transfection/luciferase assay set-up we employed and that we only carried this out without technical replicates for the primary screen. Given we utilized a highly stable and secreted luciferase (Gaussia luciferase) in a plate reader with 10 plates back-to-back, we expected – and were prepared for – false positive and false negative errors resulting from poor transfection or even contamination and carryover during pipetting. This is the reason we used a relatively liberal reduction as our cut-off and then employed several validation approaches with replicates in our experimental plan downstream. It is important to note that our approach led to the identification of several reproducible inhibitors of HCV replication. However, it is likely that we will have missed some inhibitors using this approach. One further reason for reinforcing this effect size is that to aid visualization of our data, in our initial screening, any ISGs that gave a result below background were allocated a value randomly between 0.5 and 0.01. These data are now shown in the supplementary data file 1, and we have added a limit of detection line to our figure, set at 0.5.

4)    Fig. 2E: “replication” (RLU after 72h) comprises translation *and* replication fitness, but presumably coupled in a highly non-linear fashion. Hence, I am unsure whether the ratio WT/GND is straightforward to interpret. As its results also do not impact on the further analyses in the manuscript, I recommend removing it or moving it into the supplements.

While we agree with the reviewer on the challenges in interpreting this approach, we used the ratio of WT:GND RLU to exclude CDKN1A for downstream analysis and thus we will need to retain it in the manuscript but have added a statement explicitly discussing the challenge in interpretation.

.     “Although we used WT:GND to exclude CDKN1A, we note that the biological significance of this ratio is challenging to interpret.”

5)    The authors should assemble a summarizing graph of all ISG candidates across all screening/validation experiments, e.g. as a heatmap showing the effect size of each ISG (coded as color) across all experiments. Like this, one might be able to identify further interesting hits (e.g. highly consistent but not overly huge effect size; contrary: large effect size in single assay, but none or even reversed effect in other assays).

We thank the reviewer for this excellent suggestion. We have added a supplementary table (supplementary table 2) compiling all the results following the miniscreen validation and have added a statement in the discussion regarding other hits that could be followed up in the future, such as TRIM34, SLC15A3 and FAM46C, which we do think are worthy of further study considering their impact on HCV SGR replication.

“Future work may however interrogate the role of TRIM34, FAM46C and SLC15A3 in inhibiting HCV.”

6)    In general, there is very little information on the applied statistics– I recommend adding a short paragraph to materials and methods. I understand hardly any experiments have been repeated independently, but rather only technical replicates were employed? This must be stated clearly. Fig. 4 lacks a statement on replicates and statistics. Fig. 5 uses 9 and 5 replicates (technical or biological?) for the two viruses– this is suboptimal when using SEM (as n influences SEM). Fig. 5 might be better off with SD and/or showing individual measurements on top; also a t-test could be applied, especially to ARHGEF3 in ZIKV (also: why do empty controls have no error bar– please indicate variation!). 

We have now included a statement on statistics and reproducibility, updated the figure legends with regard to nature of replicates throughout the manuscript, and have updated the Figure 5 graphs with raw RLU with mean +/- SD plus statistical analyses.

“Statistical analyses and reproducibility

We employed extensive reproduction and validation experiments to identify ARHGEF3 as an inhibitor of HCV replication when over expressed, including exploiting distinct but complementary model systems and approaches. Additionally, for Fig 5, statistical analyses (unpaired Student’s T test) is used comparing data from EMPTY and ARHGEF3 WT conditions and p values indicated.”

Minor:

  • 1: indicate that Y-axis is in “%”

This has been corrected

  • Line 231: “extend to other flaviviruses…” remove “other”

This has been corrected

Reviewer 2 Report

The manuscript by Bamford et al. described the finding of ARHGEF3/XPLN from both human and rhesus macaque as an antiviral gene against HCV, YFV, and ZIKV, by ISG screening of HCV SGR replicon cells. The XPLN was not previously shown to function as anti HCV factor, given its interest and novelty. The screening procedures were straightforward with some optimizations/selections to exclude other many SIGs, and finally ARHGEF3 was selected. Identification of host factor including ISGs against virus infection is necessary and important. However, the characterization its mode of action in antiviral function was very preliminary and insufficient to support the conclusion of this study. The following points suggested below may improve the quality of the study.

Major points:

1.      There were unsound results for the mechanistic study, mainly in Figure 4. This experiment was done on wild-type Huh7 cells that endogenous ARHGEF3 is expressing, and the inhibitory effect of ARHGEF3 of Hs and Mm on HCVcc infection was shown by viral RNA levels only, compared to negative control cells (arbitrarily normalized to 1). These data were not sufficient to make the conclusion as it stands. To mechanistically demonstrate the inhibitory effect of ARHGEF3 on HCVcc infection, the experimental data of the following experiments are needed:

(1)    In Figure 4A, the levels of HCVcc protein(s) (e.g. Core) and the proteins of ARHGEF3 Hs and Mm in the Huh7 cells are demonstrated by western blot.

(2)    An ARHGEF3-knockout Huh7 cells should be generated, and the complementation effect of ARHGEF3 on HCVcc infection should be performed by use of the cells with ARHGEF3-WT, ARHGEF3-KO, and ARHGEF3-KO+Complementation. The HCVcc infection should be evaluated by determinations of viral RNA, protein, and preferably also the infectivity titers of released HCVcc. The complementation of Human and Rhesus macaque ARHGEF3 should be performed separately.

2.      In Fig 4C, the effect HCV RNA replication (RLU) should be examined at least, with and without siRNA (or shRNA) of ARHGEF3 (targeting full-length and truncates/mutant) treatment, if ARHGEF3-KO HCV SGR replicon cells were not made.

3.      The mechanism of antiviral activity of ARHGEF3 was hardly explored in the study, the data for truncations DelN and N transfection were too preliminary (also the amino acids positions of both truncations were not clear), which severely impaired the quality of the study and conclusions. As minimum, the authors should demonstrate whether down-stream RhoGEFs effect and mTORC2 inhibition (or which activity) mediated by ARHGEF3 were required for the inhibitory effect on HCVcc infection by measuring the activation of Rho-family GTPases and interacting with mTORC2.

4.      Fig. 5. The infections of YFV and ZIKV (RLU units) should be evaluated in the cells with ARHGEF3-WT, ARHGEF3-KO, and ARHGEF3-KO+Complementations.

Minor points:

1.        In Fig. 2, Results of human and Rhesus macaque ISGs should be indicated in the figure panels.

2.        The homology and functional domains of Hs and Mm ARHGEF3, truncations, should be depicted in a figure, perhaps in Fig. 4. Aa positions for DelN and N must be given.

Author Response

Dear Editor,

Thank you for your patience with our response to reviewers given the number of points to address, and other circumstances outside of our control that prevented us from getting this to you sooner. We wish you thank you and the three kind reviewers for their helpful comments and insights into our manuscript. Firstly, we note that all three reviewers said: the work was solid and technically sound (r1), our work had interest and novelty (r2), and it was well-conducted, technically sound and convincing (r3). There were suggestions around further significant mechanistic studies and follow up experiments. However, given the time limit for resubmission and the significant amount of work suggested to be carried out, we have decided to amend our conclusions and have now altered our text, incorporating the reviewers suggestions as critical further work. We have now addressed all the comments and we hope that the manuscript is satisfactory to publish in your journal and special issue on HCV.

The manuscript by Bamford et al. described the finding of ARHGEF3/XPLN from both human and rhesus macaque as an antiviral gene against HCV, YFV, and ZIKV, by ISG screening of HCV SGR replicon cells. The XPLN was not previously shown to function as anti HCV factor, given its interest and novelty. The screening procedures were straightforward with some optimizations/selections to exclude other many SIGs, and finally ARHGEF3 was selected. Identification of host factor including ISGs against virus infection is necessary and important. However, the characterization its mode of action in antiviral function was very preliminary and insufficient to support the conclusion of this study. The following points suggested below may improve the quality of the study.

Major points:

  1. There were unsound results for the mechanistic study, mainly in Figure 4. This experiment was done on wild-type Huh7 cells that endogenous ARHGEF3 is expressing, and the inhibitory effect of ARHGEF3 of Hs and Mm on HCVcc infection was shown by viral RNA levels only, compared to negative control cells (arbitrarily normalized to 1). These data were not sufficient to make the conclusion as it stands. To mechanistically demonstrate the inhibitory effect of ARHGEF3 on HCVcc infection, the experimental data of the following experiments are needed: 

(1)  In Figure 4A, the levels of HCVcc protein(s) (e.g. Core) and the proteins of ARHGEF3 Hs and Mm in the Huh7 cells are demonstrated by western blot.

(2)    An ARHGEF3-knockout Huh7 cells should be generated, and the complementation effect of ARHGEF3 on HCVcc infection should be performed by use of the cells with ARHGEF3-WT, ARHGEF3-KO, and ARHGEF3-KO+Complementation. The HCVcc infection should be evaluated by determinations of viral RNA, protein, and preferably also the infectivity titers of released HCVcc. The complementation of Human and Rhesus macaque ARHGEF3 should be performed separately. 

  1. In Fig 4C, the effect HCV RNA replication (RLU) should be examined at least, with and without siRNA (or shRNA) of ARHGEF3 (targeting full-length and truncates/mutant) treatment, if ARHGEF3-KO HCV SGR replicon cells were not made. 
  2. The mechanism of antiviral activity of ARHGEF3 was hardly explored in the study, the data for truncations DelN and N transfection were too preliminary (also the amino acids positions of both truncations were not clear), which severely impaired the quality of the study and conclusions. As minimum, the authors should demonstrate whether down-stream RhoGEFs effect and mTORC2 inhibition (or which activity) mediated by ARHGEF3 were required for the inhibitory effect on HCVcc infection by measuring the activation of Rho-family GTPasesand interacting with mTORC2. 
  3. Fig. 5. The infections of YFV and ZIKV (RLU units) should be evaluated in the cells with ARHGEF3-WT, ARHGEF3-KO, and ARHGEF3-KO+Complementations. 

We thank the reviewer for their comments about our manuscript and their interest in the mechanistic basis of ARHGEF3 function. We agree that a limitation of our current study lies in the lack of mechanistic insight provided as to how ARHGEF3 inhibits HCV replication. We agree that the experiments very clearly laid out by the reviewer would be the ideal experiments used to assess the role of ARHGEF3 in HCV replication. However, given that these experiments would take at least 6 months for us to achieve, we believe this mechanistic insight to be outside of the scope of our current study that aimed primarily to identify ISG inhibitors of HCV replication from two species, humans and macaques. Although, the presentation of our results has placed emphasis on ARHGEF3/XPLN, as such, we have now toned-down conclusions in the title and abstract, made clear our aims and objectives, and added the reviewer’s mechanistic insight into the discussion which sets out how we would proceed in future studies. We will take the reviewers comments on-board constructively for studies in the future that would explore in detail the mechanism underlying ARHGEF3/XPLN inhibition of HCV replication.

Minor points:

  1. In Fig. 2, Results of human and Rhesus macaque ISGs should be indicated in the figure panels.

We have added this in the X axis label.

  1. The homology and functional domains of Hs and Mm ARHGEF3, truncations, should be depicted in a figure, perhaps in Fig. 4. Aa positions for DelN and N must be given.

We have added this to the text, specifically the mention that the N-terminal domain refers to positions 1-125.

Reviewer 3 Report

In this manuscript, Bamford and colleagues performed an overexpression screen of over 800 reported interferon-stimulated genes from human and monkey species to identify novel regulator of hepatitis C virus (HCV) replication. Following extensive validation with various secondary screening approaches, the authors have identified the host protein ARHGEF3 as a novel negative regulator of the viral RNA synthesis step of HCV life cycle.

Overall, the study was well conducted and technically sound, and the resulting conclusions are convincing. Notably, I have appreciated that the authors identified important determinants of ARHGEF3 and especially, that they extended their study to the clinical level by analyzing patient liver biopsies. However, as described below, I felt that some basic characterization of the mode-of-action of ARHGEF3 in inhibiting HCV replication was missing and would significantly strengthen the study.

While the antiviral role of ARHGEF3 is convincing, how this is achieved remains elusive. I acknowledge that this is challenging to address. However, imaging ARHGEF3 in infected cell by confocal microscopy might give hints about whether ARHGEF3 directly target replication complexes. What is the localization of ARHGEF3 in infected or sub-genomic replicon-containing cells? Does it change and/or does it overlap with viral proteins and/or dsRNA? Ideally, this could be done by detecting (ideally) endogenous ARHGEF3 (if an antibody is available) or Myc-tagged ARHGEF3. If a relocalization of ARHGEF3 is observed, it would be relevant to include the inactive ARHGEF3 deletion mutants in such analysis. Finally, in the discussion, the authors must elaborate in more details about the putative mechanisms underlying ARHGEF3 anti-Flaviviridae activity (line 334). The reported functions of this host factors (e.g. autophagy) are discussed but the authors do not discuss how this might participate to ARHGEF3 inhibition of HCV replication, even if this is speculative at that stage.

In addition, most of the study (if not all) was done in an overexpression set-up and did not address the role of endogenous ARHGEF3. This is especially relevant since basal expression of the protein appears to be detected in biopsies of both uninfected and HCV-infected livers (Fig 6). Fig S4 also implies that some specific signal was detected in Huh7 cell in unstimulated conditions if this is not background. In that case, the authors must address the impact of endogenous ARHGEF3 on HCV replication by decreasing its expression using RNA interference or CRISPR-Cas9 KO approaches. In the same line of ideas, both figures actually question the fact that ARHGEF3 is a bona fide ISG (at least in the context of the liver). I think that it would be relevant that the authors discuss this, including taking into consideration previous studies having convincingly demonstrated that ARHGEF3 is induced by interferons.

The authors evaluate the efficiency HCV RNA translation by measuring luciferase activity produced from a replication-defective RNA sub-genome. This was done at 72 hours post-electroporation, which is very late given the viral RNA half-life. I was actually surprised that quite bioluminescence levels were detectable at this time point (3x10e5 RLU) since the viral RNA is expected to be totally degraded at that time. Typically, assessing HCV RNA translation must be assessed shortly after transfection, i.e typically 4 hours post-electroporation.  

Other comments:

- Throughout the manuscript, the authors imply that ZIKV and YFV are hepacivruses since described as “related viruses” (Lines 33, 78, 231, 342). Yet, in the title, they wrote “other flaviviruses” in the title as if HCV was a flavivirus. It isn’t and this is misleading. The word Flaviviridae is instead much more appropriate in the title. HCV belongs to the Hepacivirus genus and ZIKV and YFV to the Flavivirus genus within the Flaviviridae family. Moreover, sentences in lines 33, 78, 231, 342 must be rephrased accordingly and it must be clearly mentioned that both ZIKV and YFV belong to a different genus in the introduction.

- Out of the 870 genes tested, how many were from each species? What is the orthologue overlap in both libraries (i.e. present in both libraries or unique) ? The same applies for the results of the screen. A Venn diagram might help to represent the results.

- The screenings were performed in human cells. Is it possible that some monkey proteins are simply inactive in human cells despite that they would exhibit a potent anti-HCV activity in monkeys? I think that this should be at least mentioned as a possibility for generating false negatives in the screen.

- Please indicate the exact names of the ZIKV and YFV strains that were used even if a reference is included.

- I would suggest to tune down the conclusions about the inactivity of N mutant since it is much less expressed than the other overexpressed forms of ARHGEF3.

- The presentation of Fig S3 is not appropriate. The molecular weight markers are not indicated and specific bands in irrelevant areas of the membrane are shown. These are the raw full-size western blot data (which should be included in a separate file).

- Line 217, this is not a KO but a point mutation

- Line 13, the verb “permissive” refers to the cells and not the virus.

- Line 73: “or indeed….”. I believe that a verb is missing here. Please rephrase.

Author Response

Dear Editor,

Thank you for your patience with our response to reviewers given the number of points to address, and other circumstances outside of our control that prevented us from getting this to you sooner. We wish you thank you and the three kind reviewers for their helpful comments and insights into our manuscript. Firstly, we note that all three reviewers said: the work was solid and technically sound (r1), our work had interest and novelty (r2), and it was well-conducted, technically sound and convincing (r3). There were suggestions around further significant mechanistic studies and follow up experiments. However, given the time limit for resubmission and the significant amount of work suggested to be carried out, we have decided to amend our conclusions and have now altered our text, incorporating the reviewers suggestions as critical further work. We have now addressed all the comments and we hope that the manuscript is satisfactory to publish in your journal and special issue on HCV.

In this manuscript, Bamford and colleagues performed an overexpression screen of over 800 reported interferon-stimulated genes from human and monkey species to identify novel regulator of hepatitis C virus (HCV) replication. Following extensive validation with various secondary screening approaches, the authors have identified the host protein ARHGEF3 as a novel negative regulator of the viral RNA synthesis step of HCV life cycle.

Overall, the study was well conducted and technically sound, and the resulting conclusions are convincing. Notably, I have appreciated that the authors identified important determinants of ARHGEF3 and especially, that they extended their study to the clinical level by analyzing patient liver biopsies. However, as described below, I felt that some basic characterization of the mode-of-action of ARHGEF3 in inhibiting HCV replication was missing and would significantly strengthen the study. 

While the antiviral role of ARHGEF3 is convincing, how this is achieved remains elusive. I acknowledge that this is challenging to address. However, imaging ARHGEF3 in infected cell by confocal microscopy might give hints about whether ARHGEF3 directly target replication complexes. What is the localization of ARHGEF3 in infected or sub-genomic replicon-containing cells? Does it change and/or does it overlap with viral proteins and/or dsRNA? Ideally, this could be done by detecting (ideally) endogenous ARHGEF3 (if an antibody is available) or Myc-tagged ARHGEF3. If a relocalization of ARHGEF3 is observed, it would be relevant to include the inactive ARHGEF3 deletion mutants in such analysis. Finally, in the discussion, the authors must elaborate in more details about the putative mechanisms underlying ARHGEF3 anti-Flaviviridae activity (line 334). The reported functions of this host factors (e.g. autophagy) are discussed but the authors do not discuss how this might participate to ARHGEF3 inhibition of HCV replication, even if this is speculative at that stage. 

In addition, most of the study (if not all) was done in an overexpression set-up and did not address the role of endogenous ARHGEF3. This is especially relevant since basal expression of the protein appears to be detected in biopsies of both uninfected and HCV-infected livers (Fig 6). Fig S4 also implies that some specific signal was detected in Huh7 cell in unstimulated conditions if this is not background. In that case, the authors must address the impact of endogenous ARHGEF3 on HCV replication by decreasing its expression using RNA interference or CRISPR-Cas9 KO approaches. In the same line of ideas, both figures actually question the fact that ARHGEF3 is a bona fide ISG (at least in the context of the liver). I think that it would be relevant that the authors discuss this, including taking into consideration previous studies having convincingly demonstrated that ARHGEF3 is induced by interferons.

We agree that the antiviral capabilities of over-expressed ARHGEF3 is convincing and that one limitation of our work is the lack of mechanistic studies. The concerns of reviewer 3 are similar to those of reviewer 2 and we appreciate their helpful and constructive advice on how to explore this, which will undoubtedly shape our plans for future studies.

We have added additional text discussing potential speculative mechanisms through which ARHGEF3 may reduce HCV replication.

Speculatively, it is possible that ARHGEF3 directly targets early replication processes shared by members of the Flaviviridae, through limiting autophagy, which is needed for the formation of replication organelles of these viruses (as reviewed in https://www.ncbi.nlm.nih.gov/pmc/articles/PMC6321027/#:~:text=In%20recent%20years%2C%20emerging%20lines,damage%20%5B16%2C17%5D).

We agree with the reviewer that one experiment to follow up on would be to explore the role of endogenous ARHGEF3/XPLN in HCV infection, as we do detect clear but low-level expression of ARHGEF3 in liver cells in vitro but also within the liver of infected patients in vivo. While we admit that our results of viral/IFN-mediated induction of ARHGEF3 in vitro and in vivo are very modest (<2-fold), other studies have shown ARHGEF3/XPLN as an ISG in both humans (using interferome database and see: https://pubmed.ncbi.nlm.nih.gov/27357150/)  and in other animals (http://isg.data.cvr.ac.uk/) . However, these studies were not carried out in liver cells but in PBMCs and in fibroblasts. Further work should explore ARHGEF3 in other virus/cell systems.

The authors evaluate the efficiency HCV RNA translation by measuring luciferase activity produced from a replication-defective RNA sub-genome. This was done at 72 hours post-electroporation, which is very late given the viral RNA half-life. I was actually surprised that quite bioluminescence levels were detectable at this time point (3x10e5 RLU) since the viral RNA is expected to be totally degraded at that time. Typically, assessing HCV RNA translation must be assessed shortly after transfection, i.e typically 4 hours post-electroporation.  

This is very likely because the system we employed uses Gaussia luciferase, which is secreted, accumulates in the extracellular space, and is highly stable compared to the traditional luciferases used in HCV replicon studies like firefly luciferase. Given our own experience with Gaussian luciferase, the values we obtained at 72 hours post electroporation for the replication-defective SGR are typical. 

Other comments:

- Throughout the manuscript, the authors imply that ZIKV and YFV are hepacivruses since described as “related viruses” (Lines 33, 78, 231, 342). Yet, in the title, they wrote “other flaviviruses” in the title as if HCV was a flavivirus. It isn’t and this is misleading. The word Flaviviridae is instead much more appropriate in the title. HCV belongs to the Hepacivirus genus and ZIKV and YFV to the Flavivirus genus within the Flaviviridae family. Moreover, sentences in lines 33, 78, 231, 342 must be rephrased accordingly and it must be clearly mentioned that both ZIKV and YFV belong to a different genus in the introduction.

We thank the reviewer for these taxonomic points and we agree that as we wrote it is could be misleading (although this was not our intent) and we have changed the title and text in each of the cases to make it explicitly clear of the taxonomic relationships of HCV and YFV/ZIKV.

- Out of the 870 genes tested, how many were from each species? What is the orthologue overlap in both libraries (i.e. present in both libraries or unique) ? The same applies for the results of the screen. A Venn diagram might help to represent the results.

We have added this into the text during the description of the libraries used. Specifically, this combined library contains 752 independent ISGs, including 493 unique ISGs (275 shared, 252 human-specific, and 69 macaque-specific).

- The screenings were performed in human cells. Is it possible that some monkey proteins are simply inactive in human cells despite that they would exhibit a potent anti-HCV activity in monkeys? I think that this should be at least mentioned as a possibility for generating false negatives in the screen. 

We have added the following additional sentence to outline that activity of ISGs could both positively and negatively be affected by host cell species.

One consideration for our approach is that the antiviral activity of ISGs may be dependent on the host cell and thus it is possible that expression of macaque ISGs in human cells could enhance or reduce their activity.

- Please indicate the exact names of the ZIKV and YFV strains that were used is now included.

We have included these specifically in the methods section as well as the reference that was included. The strains are: ZIKV: a recombinant Asian genotype BeH819015 with UTRs from PE243 (https://pubmed.ncbi.nlm.nih.gov/29022864/ ) ( and YFV: a recombinant vaccine 17D (https://pubmed.ncbi.nlm.nih.gov/32980446/).

- I would suggest to tune down the conclusions about the inactivity of N mutant since it is much less expressed than the other overexpressed forms of ARHGEF3.

We have adjusted our conclusions and added a sentence:

However, care must be taken in interpreting results from the N domain given its reduced expression at the protein level.

- The presentation of Fig S3 is not appropriate. The molecular weight markers are not indicated and specific bands in irrelevant areas of the membrane are shown. These are the raw full-size western blot data (which should be included in a separate file). 

We have added in MW for the markers. We felt that given this is a supplementary piece of data it was better to include the full blot for the sake of transparency.

- Line 217, this is not a KO but a point mutation

We have removed reference to KO and inserted:

To test whether either of these functions contributed to the antiviral activity of ARHGEF3, we made a single RhoGEF activity ablation point mutant (W440L hereafter termed 'delGEF’),...”.

- Line 13, the verb “permissive” refers to the cells and not the virus.

This has been changed.

- Line 73: “or indeed….”. I believe that a verb is missing here. Please rephrase.#

We have removed the word indeed.

Round 2

Reviewer 3 Report

The authors have answered all my comments and I accept the paper as is.